# Motion Recognition Method for Construction Workers Using Selective Depth Inspection and Optimal Inertial Measurement Unit Sensors

**Tingsong Chen** *[ID], **Nobuyoshi Yabuki** [ID] **and Tomohiro Fukuda** [ID]

Graduate School of Engineering, Osaka University, Osaka 565-0871, Japan
* Correspondence: chen.tingsong@it.see.eng.osaka-u.ac.jp

**Abstract:** The construction industry holds the worst safety record compared to other industrial sectors, and approximately 88% of accidents result in worker injury. Meanwhile, after the development and wide application of deep learning in recent years, image processing has greatly improved the accuracy of human motion detection. However, owing to equipment limitations, it is difficult to effectively improve depth-related problems. Wearable devices have also become popular recently, but because construction workers generally work outdoors, the variable environment makes the application of wearable devices more difficult. Therefore, reducing the burden on workers while stabilizing the detection accuracy is also an issue that needs to be considered. In this paper, an integrated sensor fusion method is proposed for the hazard prevention of construction workers. First, a new approach, called selective depth inspection (SDI), was proposed. This approach adds preprocessing and imaging assistance to the ordinary depth map optimization, thereby significantly improving the calculation efficiency and accuracy. Second, a multi-sensor-based motion recognition system for construction sites was proposed, which combines different kinds of signals to analyze and correct the movement of workers on the site, to improve the detection accuracy and efficiency of the specific body motions at construction sites.

**Keywords:** sensor fusion; motion recognition; hazard prevention; inertial measurement unit; depth camera



## 1. Introduction

The concept of a "Smart City" has been widely known by people around the world for many years. Its main purpose is to use information technology to help to improve city services. Currently, the smart city concept is being applied in transportation, citizen management, and urban resource allocation, all with good performance compared with traditional methods. Along with the rise of the smart city, the concept of "smart construction" [1] has also been proposed recently, in which many well-applied methods from other fields are exported to the construction area. However, owing to the differences in management mode and implementation, these approaches have not performed well. Thus, there is much room for the improvement in the development of smart construction.

This lackluster performance is especially concerning in light of the construction industry field currently holding the worst record for safety compared with other industries: approximately 88% of workplace incidents in the construction industry are caused by unsafe behaviors [2]. For example, in Japan, the average number of fatalities in construction accidents is over 300 people per year [3], and this performance has not improved well during the last 20 years [4]. In comparison with other developed countries, accident monitoring efficiency in Japan is the main reason that injured workers cannot be identified and rescued in time.

After the development and wide application of deep learning in recent years, image processing has greatly improved the accuracy of human motion detection. However, single

visual sensor detection still has its unreliability, it is easily affected by changes in the surrounding environment and the location of the measured object.

Meanwhile, wearable devices have become increasingly popular recently. Some simple devices such as watches can perform basic body path or state detection. However, if these devices are the only angle of detection, they lack reliability. Complex special clothing with hundreds of detection points can perform extremely detailed tracking. However, because of its inconvenience, worker resistance, and high cost, it is not suitable for use in ordinary construction environments.

In construction sites, the main causes of worker injury and death are related to health issues such as heat stroke, physical injuries from falling, contacting, and mental injuries. Some of the injuries are caused by accidents, and some of them are caused by unsupervised unsafe acts taken by workers as a result of cost limitations, time pressure, and other reasons. Normal monitoring systems, such as web cameras, require manual operation and are inefficient because of human neglect, visual obstacles, and other factors. A more efficient and accurate safety management and monitoring method is needed.

Camera-based monitoring methods are widely used and researched around the world because of their many benefits, such as low cost and ease of assembly. However, unavoidable disadvantages are more numerous, such as low accuracy when it is too far away from the visual sensor or in poor lighting conditions. Some approaches concentrate on changing the RGB frame to an RGB-D frame by adding depth to pictures with machines such as Kinect [5] and RealSense [6]. RGB-D is more accurate for detecting humans and objects compared with RGB pixel cropping, and it also works well even under poor lighting conditions. However, owing to the limitations of working distance, reflective surfaces, and relative surface angles, depth maps in RGB-D frames always contain significant holes and noise, and these errors limit the practical use of RGB-D frames in real applications. Thus, depth maps for filling holes and removing noise are necessary steps in depth camera-based monitoring systems. Because of the high error rate of depth measurement over wide areas and long distances, depth scanning of large areas also increases the calculation demands. Therefore, an approach that can effectively achieve an in-depth analysis of designated areas is needed.

Monitoring methods based on inertial measurement unit (IMU) sensors have also been gaining attention in recent years because of their clear benefits compared with other methods, such as those relying on visual cameras. IMU sensors are non-intrusive, lightweight, and portable measuring devices that, when attached to a subject, can overcome the sensor viewpoint to detect activities in a non-hindering manner [7–9]. After preprocessing of the motion for recognition, discriminative features are then derived from time and/or frequency domain representations of the motion signals [10] and used for activity classification [11]. Although there are many benefits of IMU sensors, there are also some disadvantages. First, their output is not intuitively understood and not amenable to manual rechecks. Second, model complexity is hard to control, especially when precise motion capture is needed. Generally, construction workers stay outdoors, and the changing environment there makes them more resistant to accepting a huge number of wearable devices, such as some complex special clothing with hundreds of detection points on it. Therefore, reducing the burden on workers while stabilizing detection accuracy is an issue that needs to be resolved.

The construction site is a complex environment which includes many corners, blind spots, and isolated spaces. If only one kind of sensor (such as a visual sensor) is used, due to the effect of lighting, shooting angles, etc., the detection accuracy can be largely changed. For example, once the object or human is outside the optimal recognition area of the visual sensor, auxiliary recognition from other kinds of the sensor will be needed to ensure efficiency. In conclusion, a multi-sensor-based recognition method is needed.

To solve the problems discussed above, we propose an advanced hazard prevention method for workers based on multiple sensor networks. This approach aims at using worker data obtained from different kinds of sensors to detect human motion and move-

ment more accurately, including RGB cameras, depth cameras, and IMUs, and combining them cooperatively to increase the detection accuracy of special motions of workers, and improve efficiency for accident alert and injury rescue.

To address the research gap, the purpose of this study was to examine the application of deep learning to the labor-intensive and time-consuming removal of unwanted features in images of repetitive infrastructure. The effect of unwanted-feature removal on the structure-from-motion process was also investigated. First, a new method, called selective depth inspection (SDI), was proposed. This approach adds preprocessing and imaging assistance to the ordinary depth map optimization, thereby significantly improving the calculation efficiency and accuracy. Second, a multi-sensor-based motion recognition system for construction sites was proposed, which combines different kinds of signals to analyze and correct the movement of workers on the site, to improve the detection accuracy and efficiency of the specific body motions at construction sites.

The remaining sections are organized as follows. A literature review is provided in Section 2, including camera-based, depth camera-based, and IMU-based methods. The proposed methodology is introduced in Section 3, where visual sensor- and IMU-based human recognition and the sensor fusion approach are introduced. The simulation and experiment details and hardware parameters are introduced in Section 4. Discussion of this research is given in Section 5. Finally, conclusions are drawn in Section 6.

## 2. Literature Review

### 2.1. RGB Camera-Based Human Modeling

RGB camera-based motion recognition is inexpensive and can be widely distributed, but because of its photosensitivity, its applicable environment is narrow and easily affected [12].

Commercial camera-based human detection systems require subjects to wear markers and depend on multiple cameras' calibration around the environment [13], which is inconvenient. To overcome these constraints, other researchers have focused on developing markerless approaches using multiple cameras, but some of these methods require offline processing to obtain high-quality results [14,15]. While other real-time approaches have been proposed [16], these approaches often combine a skeletal model with image data. Other real-time approaches include combining generative and discriminative methods [17]. However, multi-view methods require stationary and well-calibrated cameras, and therefore, they are not suitable for mobile scenarios.

### 2.2. Depth Camera-Based Depth Map Restoration and Human Modeling

Most motion recognition based on depth cameras is performed at a short distance, so its range of practical application is relatively narrow and cannot be easily expanded [18,19].

As for regular cameras, RGB-based depth prediction normally relies on a large body of literature and is trained with ground truth data only [20–24]. As for depth cameras, many approaches have been proposed to restore depth maps with Kinect; among these methods, two types can be divided: methods that are based on filtering and methods that are based on reconstruction. Approaches based on filtering use different kinds of filters to restore captured depth maps. A median filter is proposed to recursively fill holes in a depth map [25]. However, the sharpness of the edges is too blurred. To keep sharp edges, a joint bilateral filter is iteratively applied in a depth map [26]. Ref. [27] considered using temporal information to restore the depth map, but this approach incurs a delay because it uses multiple consecutive frames for the restoration. Methods that are based on reconstruction use image inpainting to fill missing values in depth maps. The fast-marching method (FMM) is proposed for depth value reconstruction [28]. A texture-assisted approach is also proposed, in which the texture edge information is extracted to help the restoration of the depth value [29]. These methods can remove noise and fill small size holes in in-depth maps, but when it comes to big holes in in-depth maps, the results are unsatisfactory.

To address human modeling, a method named shape completion and animation of people (SCAPE) is proposed [30], which is a data-driven approach to 3D human model

building from both shape and pose aspects. They showed that, with a high-resolution depth image from a single viewpoint, the SCAPE model can obtain the observation data. Based on the SCAPE-parameterized model, Ref. [31] combined low-resolution scans and views of a person from different angles to construct an accurate human 3D model. Ref. [32] proposed a monocular depth camera 3D human modeling approach based on referring to some previous human body pose and shape approaches that it mentioned.

### 2.3. IMU-Based Human Modeling

Roetenberg et al. [33] used 17 IMUs with the function of magnetometers, gyroscopes, and 3D accelerometers, which were fused by a Kalman filter. Supposing the measurements are with no noise and no drift, the 17 IMUs can define the full pose of the subject (in standard skeletal models). However, 17 IMUs are too numerous and very intrusive for the subject. Problems and inconveniences, such as long setup times and placing a sensor in the wrong position, commonly occur, which makes this approach difficult to reproduce.

Marcard et al. [34] computed accurate 3D poses with only six IMUs. They placed synthetic IMU sensors on a skinned multi-person linear body model in a generic way. They solved for the sequence of body poses that matched the observed actual measured sequence by optimizing the entire sequence [35]. However, this approach relies on computationally expensive offline optimization, which is also hard to reproduce.

Wearing a large number of sensors is not suitable for long-term outdoor work and increases the computing burden [36]. Therefore, a smaller number of IMU sensors and less computation complexity is key to IMU-based human modeling and motion detection.

A body pose recognition system based on IMU sensors for upper body identification of workshop workers was proposed by [37]. To obtain more accurate motion data, two IMU sensors were, respectively, placed on the experimenter's arms, one on the neck, and one on the waist (a total of 6 sensors) to achieve pose recognition for fixed movements. Although a high level of precision was achieved, including detection of posture angles, the system only supported two movements, standing, and bending.

Ref. [38] implemented motion command analysis for an electric wheeled walker using only one IMU sensor, which analyzed acceleration and drove emergency behaviors such as electric wheel braking to ensure the safety of the carrier. However, the study still had significant limitations in terms of application object and diversity of motion.

Ref. [39] proposed a fatigue testing framework based on the construction industry using a single IMU and forearm electromyography. Since the primary evaluation indicators were physiological information, the application was relatively narrow. The authors of [40] identified construction workers' activities using information from three IMU sensors and achieved high recognition accuracy. However, due to the placement of sensors only on one arm, there were limitations in recognizing some complex limb movements.

## 3. Proposed Methodology

### 3.1. Overview

In this paper, we propose an integrated sensor network method for the hazard prevention of construction workers. As shown in Figure 1, this approach mainly uses a depth camera and IMU sensors to collect data from workers and construct a human model to analyze motions and gestures. In the visual-based motion recognition step, we use image-processing-based real-time monitoring as a preprocessing step, to improve the accuracy of depth optimization. In the IMU-based motion recognition step, depending on the application environment, the number of wearable devices is reduced to a minimum to increase practicality while maintaining the detection accuracy of basic and relatively complex actions. This method concentrates on multi-sensor cooperation using different kinds of sensors to decrease the errors caused by sensor defects increase the detection accuracy and improve efficiency.

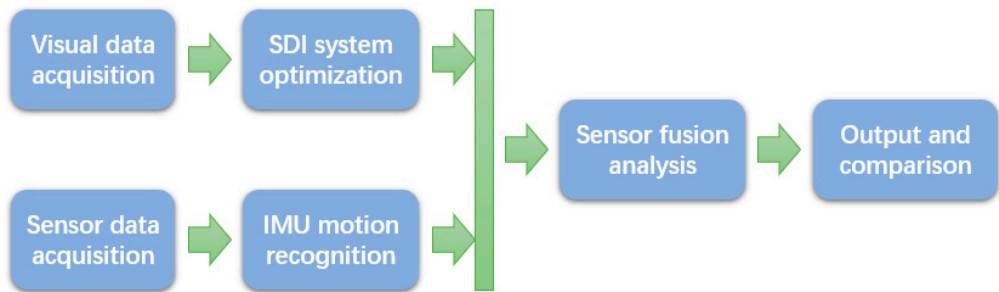

**Figure 1.** Overview of the proposed methodology.

### 3.2. Human Recognition by SDI

In typical depth detection, the depth camera has a high error rate over a long distance; for example, for RealSense D435i, the root means square (RMS) error increases when the object is 4 m or farther away from the camera. Meanwhile, the long-distance depth analysis will significantly increase the calculation burden, and the result of depth detection may have a high error rate. The proposed SDI method can divide depth detection into two steps. The first step is to use image processing to identify the monitoring area to find out whether it contains humans. The second step is to perform depth recognition and optimization in the selected areas. Preprocessing can effectively reduce the computational burden during depth optimization. At the same time, owing to depth optimization, the depth detection of a specific area can increase the effective recognition distance, to achieve longer-distance detection and reduce recognition errors caused by specific actions.

Regarding the preprocessing stage of the selected area, we use a variety of image recognition methods. The first is human recognition based on TensorFlow Lite [41]. This method can quickly identify a possible human shape in the scene through real-time analysis. Based on different training sets, the accuracy of human body recognition is also different. Another approach is skeleton recognition by PoseNet, which is also based on TensorFlow Lite [42]. This method can perform rough bone recognition through an RGB camera, thereby improving the detection accuracy during depth analysis and optimization. In our proposal, we use a microcomputer as our terminal, to ensure recognition efficiency; the above models are adopted, and though they are a little outdated, they can meet our experimental requirements of a low-consumption terminal, and at the same time, they can provide an acceptable recognition fluency.

When a human signal appears in the monitoring range, the RGB camera will detect the body frame, and only the depth data inside the selected area will be collected. The raw depth information collected may cause too many holes and defects owing to factors such as distance and environmental interference. To fix the hole problem and other interference, depth information optimization is needed at this time. This optimization can fill in the missing depth information at a long distance to effectively increase the effective distance of depth motion recognition.

Yin et al. [43] proposed a two-stage stacked hourglass network based on Varol et al. [44] to obtain high-quality results for human depth prediction. Instead of using RGB images directly, this approach uses RGB images and human part segmentation together to predict human depth. It consists of convolution layers, a part-segmentation module, and a depth prediction module. First, the RGB image input goes through the convolution layer and is converted into heat maps, after which it enters the part-segmentation module. Then, the heat maps are converted into human part-segmentation results, and these heat maps are summed as the input of the following depth prediction module with the features of previous layers. Finally, human depth prediction results are output.

Algorithm 1 above is called Gradient Fast-Marching Method (GradientFMM) [43], and it propagates the depth from known pixels to unknown pixels. After the process, every pixel in the unknown region of a depth map will be assigned a depth value. In this study, to

extend the detectable distance of the selected area, the GradientFMM algorithm is applied for depth information optimization.

---

**Algorithm 1:** Gradient Fast-Marching Method (GradientFMM)

1.   **Procedure** GradientFMM (*depthmap*)
2.   *Known* ← all pixels with known values in *depthmap*
3.   *Unknown* ← all unknown pixels adjacent to *Known* in *depthmap*
4.   insert all pixels in *Unknown* into min-heap
5.   **while** *Unknown* not empty **do**
6.       *p* ← root of min-heap
7.       calculate *p* values using *depth value equation*
8.       add *p* to *Known*
9.       remove *p* from *Unknown*
10.     perform down heap
11.     **for** each neighbor *q* of *A* **do**
12.         **if** *q* not in *Known* and *Unknown* **then**
13.             add *q* to *Unknown*
14.             perform up heap
15.         **end if**
16.     **end for**
17.   **end while**
18.   return *Known*
19. **end procedure**

---

The depth camera produces images with a resolution of $848 \times 480$ pixels at a framerate of 30 frames per second. As shown in Figure 2, in the optimization process, we first apply the GradientFMM algorithm to analyze each frame to fix undetected points of selected human areas, and then the area is considered as 3D coordinates by cooperation with the skeleton detection processing from the previous step. Finally, the depth information as a three-dimensional coordinate of each point of interest from the possible human shape can be obtained. The points of interest include but are not limited to the hands, elbows, head, waist, knees, and feet.

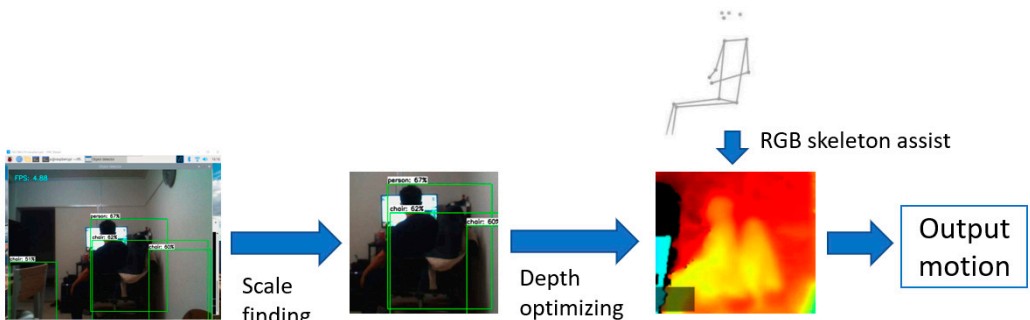

**Figure 2.** SDI method process.

*3.3. Portable Computing Terminal*

Some hazard prevention schemes are based on sensors placed on helmets, and the information collected by sensors on helmets can be analyzed only after workers end their shifts and remove their helmets. This kind of analysis method is relatively inefficient and cannot provide timely accident alerts.

In this study, as shown in Figure 3, a portable computing terminal is used to divide the processing and perform data analysis for each small sensor locally, thereby reducing the transmission of data for large images and improving processing efficiency.

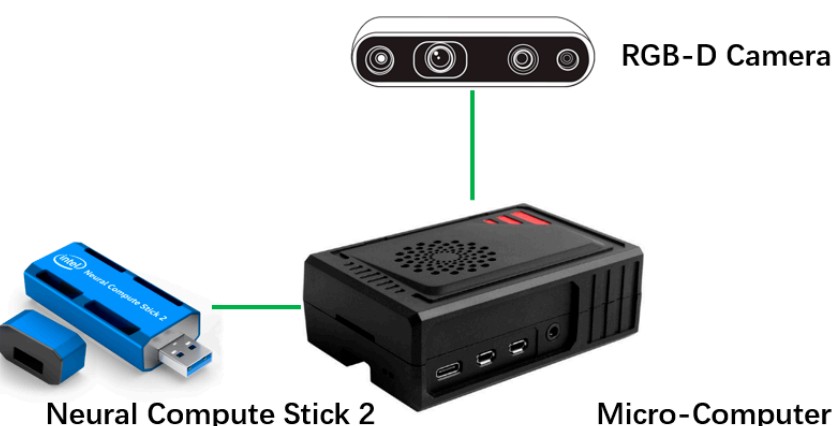

**Figure 3.** Portable computing terminal.

Recently, Raspberry Pi [45] has become one of the most popular microcomputers in the world owing to its portability, size (of a credit card), extremely low power consumption, and complete internal structure. In addition, Raspberry Pi uses the open-source Linux system as its main operating system, which makes it extremely scalable, and it can realize the required functions with the help of many open-source projects.

However, Raspberry Pi also has some shortcomings. For example, because of its low power consumption, it cannot drive some large GPUs for deep learning training, and the integrated graphics card that it carries cannot do this work as well. Thus, in this research, the Neural Compute Stick 2 (NCS2) produced by Intel was used as an external neural network accelerator to make up for this shortcoming. This device can accelerate neural network inference operations at relatively low operating power. NCS2 needs to be used in conjunction with OpenVINO [46]. This is an open-source software developed by Intel, which mainly includes the Model Optimizer and an inference engine to receive neural network architecture and weight information.

Another important part is the visual information collection unit. In this study, the RealSense R435i depth camera produced by Intel was used to collect 2D image information and 3D depth information.

Compared with centralized computing, the portable computing unit has better scalability and reduces the requirements for data transmission. It can perform real-time processing at the sensor and transmit the results to the central processing unit for rapid analysis. In some large construction sites, where a large number of sensors need to be deployed, the computational burden on the central processing unit can be effectively relieved. At the same time, because of the mobile computing unit's high portability, real-time processing, and early warning capabilities, it can be deployed at the dead ends of construction sites, blind areas of large operating vehicles, and other accident-prone areas, thereby speeding up rescue after accidents and reducing secondary injuries.

### 3.4. IMU-Based Human Motion Recognition

This section introduces IMU-based human motion detection, in which IMUs measure triaxial (3D) accelerations and triaxial angular velocities. This approach can also easily obtain information directly without numerous restrictions.

In our proposal, we mainly consider motion capture during work activities, so the body's four limbs are the observation focus. The limbs can express most of the essentials of movements. Despite minimizing the number of wearable devices, we still maintain the detection accuracy of basic and relatively complex movements.

IMU motion recognition is based on Dehzangi et al. [47], who introduced a human activity recognition method in the normal environment; the activities they considered are walking, walking upstairs, walking downstairs, sitting, standing, and sleeping. In our proposal, because our subjects were construction workers, new motions are added: lifting objects (one or two arms are elevated), picking up heavy objects (the swing amplitude of

both arms is reduced and stiff), holding up heavy objects (the arms are partially angled and stiff), raising arms (arms are at right angles to body), regular cyclical movement (arms making a circular motion), bending over (leaning forward or backward), and kneeling (one or both knees).

The framework of the IMU-based human motion recognition system is as follows. First, relevant data are collected from users. Each motion is divided according to the time axis, and information such as the acceleration changes in the four sensors during the time is obtained. At the same time, for different motions, feature extraction is performed on the path changes in each sensor, and the activity label is created. Finally, by comparing the new action with the label data in the database, the action with the highest similarity is output as the result.

Each worker wears 4 sensors, and each sensor owns a separate port identification code. The combined data group from the 4 sensors will be judged as one worker. All four IMU sensors continuously record data from each worker, and the motion recognition method analyzes amplitude changes. At the same time, when a motion process changes dramatically, the differences between triaxial accelerations and angular velocities before and after the change are counted and recorded as change graphs. Finally, the differences are compared with a motion database to identify the best match. The sensors can not only collect motion data, but they can also collect information about the workers' surrounding environment, such as temperature, height, and air pressure, by exchanging data with environmental sensors to ensure that workers are in a proper working environment.

*3.5. Multi-Sensor Fusion and Analysis*

Normally, because visual signals and IMU electronic signals are quite different, it is difficult to make a comparison between them. In our proposal, both the depth camera-based method and IMU sensor-based method can obtain results independently, but when it comes to some specific circumstances, such as self-occlusion, using only one kind of signal will cause a high error rate and affect the whole system.

In this research, two kinds of signals are cooperatively used by reducing the advantages and disadvantages of each to further improve accuracy. The detection area of the depth camera is considered a huge 3D coordinate system. The depth camera is placed on one side of the system, and IMU sensors are also calibrated before loading to make sure they are consistent at time 0.

As described above, when recording starts, both the camera and IMU sides generate constant 3D coordinate changes. For the depth camera side, the variation and value of specific points are obtained from the coordinates in the depth map and frame platform. For the IMU sensor side, during movement the three axes change with different accelerations, and using the origin set at time 0, the path changes and distance are calculated by Equation (1).

$$\vec{S} = \int \left( \int \left( \vec{a} \right) dt \right) dt,$$

(1)

where $\vec{S}$ represents directional distance and $\vec{a}$ represent average acceleration during time period $t$. Although the units, distance, and size are quite different between depth map coordinates and IMU sensor coordinates, we can describe the change amplitude curve between each set of specified coordinate points (in this case, points of two elbows and two knees) by considering the weight of each kind of sensor. Thus, a more accurate result is output for comparison with our database, resulting in higher reliability for human motion recognition.

The final degree of change is shown in Equation (2).

$$\Delta P = \frac{\frac{\Delta P_v}{P_v^0} \Delta \alpha + \frac{\Delta P_I}{P_I^0} \Delta \beta}{2}$$

(2)

where $\Delta P_v$ is the change in motion from the visual side, $\Delta P_I$ is the change in motion from the IMU sensor side, $P_v^0$ and $P_I^0$ are the initial states of the current time segment, and $\alpha$ and $\beta$ are weight coefficients for visual and IMU sides, respectively.

In this study, since the effectiveness of existing sensors cannot be evaluated in advance, in this experiment, the weight distribution between different sensors is relatively ideal, and in this experiment, they are distributed into 50%, 50%.

Figure 4 shows a flow chart of the cooperative method for fusing both data streams. The key point of this research is how to compare the data of the visual sensor and IMU sensor at the same latitude. In this experiment, a concept called "degree of change" was proposed; it is described as the change amplitude curve between each set of specified coordinate points (in this case, points of two elbows and two knees) by considering the weight of 2 kinds of sensors. The recognition based on the visual sensor will mark the interest points through skeleton detection, and the pixel path in 3d coordinates can be calculated. The recognition based on the IMU sensor will collect the acceleration and angular velocity, by using the double integration method, the distance and path in 3d coordinates can be calculated. Finally, the degree of change formula is used to obtain the degree of change in each interest point, to compare with the database and find the best match.

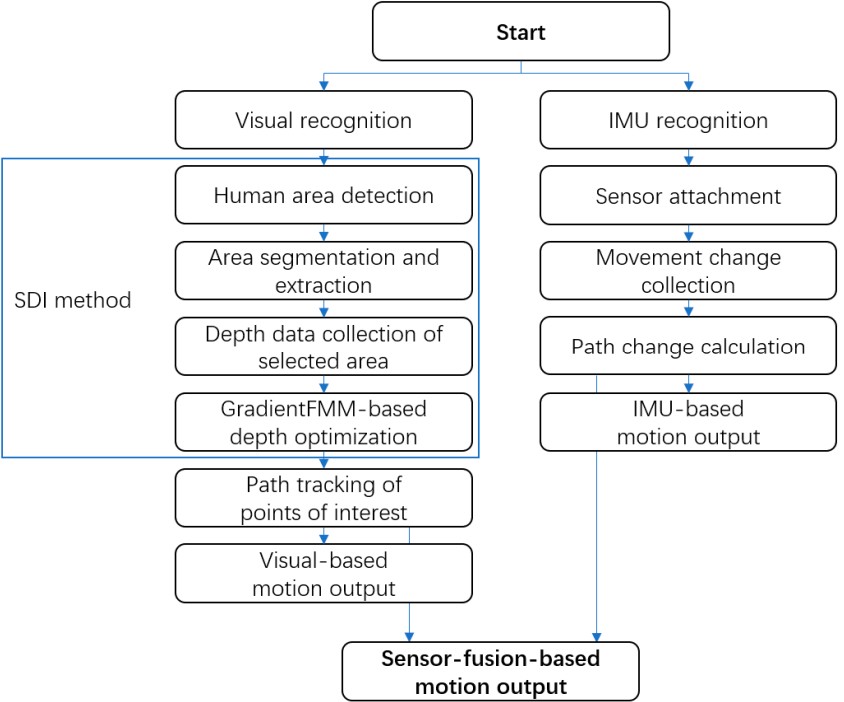

**Figure 4.** Flow chart of the proposed multi-sensor-based motion recognition.

## 4. Experimentation

This section describes a simulation to validate the feasibility of our proposal. It includes the following aspects:

(1) We use a depth camera along with the SDI method to identify possible humans in an area and detect the human skeleton, optimize the depth map of the human area, and collect the path change in interest points in a 3D coordinate environment.

(2) We use attached IMU sensors to detect the human skeleton based on four points of interest, and to record the coordinate differences to identify different motions.

(3) Our weight-based multi-signal fusion correction approach generates coordinate differences from each frame and frequency and then outputs accurate position information.

To obtain the depth maps of subjects, a portable computing terminal was constructed from a Raspberry Pi 4 as a computing unit, an NCS2 as a training accelerator, and an Intel RealSense D435i depth camera as a data collecting unit positioned at one side of our

experiment area. This camera can achieve smooth video streaming with 848 × 480 pixel resolution at 30 frames per second. The possible depth detection ranges from 0.5 to 16 m.

To obtain the 3D motion data of subjects, four WitMotion IMU sensors were attached to both elbows and knees of each subject. At the outset of this experiment, sensors were selected with the goal of detecting worker safety from multiple angles, including environmental factors such as high-temperature work (heat stroke) and high-altitude work (falling, etc.). This consideration led to a different selection of sensors. Sensors for elbows are closer to the heart and head, so models that detect altitude, temperature, barometric pressure, and location information are used. For the knees, a functionally simpler model was used. The choice of the sensor did not affect the results of this experiment. The knee sensor model was BWT901CL; this sensor supports USB and Bluetooth 2.0 as its transmission method, and the detectable distance can reach 10 m at most. The baud rate of it is 115,200 Bd, and the sampling rate of it is 60 Hz. Meanwhile, the battery life of this sensor is about 4 h, which is suitable for most workers. We used the WTGAHRS2 sensor for the elbows; it can provide a more accurate 3-axis inspection than the knee sensor; the baud rate of it is 9600 Bd, and the sampling rate of it is 30 Hz. Yet, the elbow sensor does not have a Bluetooth module; this problem can be solved by connecting to an external Bluetooth device. Currently, the diversity of collected data is balanced with its low portability; in the future, this sensor will be considered to change to a more portable kind. Each IMU will provide a three-axis acceleration detection, giving four of them together 12-dimensional detection on the object. These IMU sensors can detect the above parameters plus air pressure and elevation, which can help to verify that the environment surrounding workers is stable and comfortable. A picture of the IMU sensor architecture is shown in Figure 5.

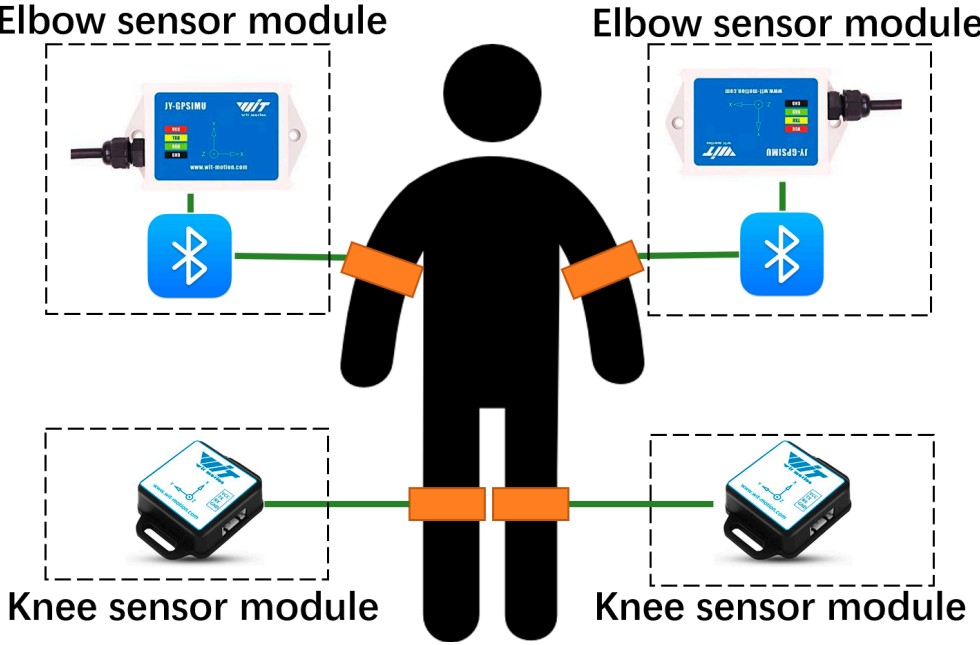

**Figure 5.** IMU sensor architecture.

In this experiment, the visual sensor is directly connected to the microcomputer via USB, to provide stable large-capacity image and video transmission. The IMU sensors are connected via Bluetooth, through the built-in and external Bluetooth module, to collect motion data in real-time. Each microcomputer can be considered as a local terminal; in the real world, environment, multiple local terminals will be connected by Wireless Local Area Network (WLAN), to provide multi-point monitoring.

Before the beginning of the experiment, a database of actions is tested, which is also considered the comparison group. In this experiment, several motions were considered, including normal motions such as standing, sitting, and lying down, and motions specific

to construction work, such as lifting objects, picking up heavy objects, holding up heavy objects, raising arms, regular cyclical arm movements, bending over, and kneeling. The reason to apply these motions is, in actual work, the possible motions of workers will be more complicated than in daily life, and many accidents also arise because of these actions (injuries to hands, waist, knees, etc.). By strengthening the monitoring and identification of the special motions, it is possible to effectively and quickly find out workers who continue to be in abnormal motion, thereby avoiding more serious accidents.

Here, we give a detailed description of the considered motions. Of these 10 motions, 3 are normal motions and 7 are specific to construction. All motions are considered as starting from facing ahead. Among the normal motions, we have standing, sitting, and sleeping (lying down), which are also commonly seen in other studies. Construction motions are motions that are commonly seen at construction sites, especially when workers are working in narrow areas or when they must reach a height that a normal standing person would find hard to reach. A mirror schematic diagram of these motions is shown in Figure 6, where the yellow circles in the figures are the points of interest where sensors are attached.

| | | | | | |
|---|---|---|---|---|---|
| **Standing** | | **Standing** is described as the arms drop naturally, swing slightly, the legs are stand and relaxed, and the body is not stiff. | **Holding up heavy object** | | **Holding up heavy objects** is described as embracing the heavy object with both hands, the body is leaning forward, and knees are slightly bent, and the body is stiff. |
| **Sitting** | | **Sitting** is described as sit down naturally, facing forward with a chair underneath, with legs at right angles, and arms hanging down naturally. | **Raising arms** | | **Raising arms** is described as stand naturally with hands raised from both sides, flush with shoulders. |
| **Lying down** | | **Lying down** is described as all body parts are touched or close to the floor and lie on the ground. | **Regular cyclical movement** | | **Regular cyclical movement** is described as lean forward slightly, fix the object with the left hand, and make a circular motion with the right elbow at the up/down and front/back angles. |
| **Lifting objects** | | **Lifting objects** is described as stand relaxed and raise hands naturally, with elbows approximately flush with position of ears. | **Bending over** | | **Bending over** is described as similar to a bowing motion, lean forward, and naturally close hands at body sides, showing a movement that is not too rigid. |
| **Picking up heavy object** | | **Picking up heavy objects** is described as lifting heavy objects with both hands. | **Kneeling** | | **Kneeling** is described as kneel naturally, with thighs still nearly straight up, facing forward, and leaning forward slightly. |

**Figure 6.** A detailed description of detected motions.

The data collecting and experiment area are shown in Figure 7. The portable computing terminal was placed at the front of the whole area, the IMU sensors were attached to the subject's elbows and knees, and then the workers made the 10 motions (Table 1) in front of the depth camera.

**Table 1.** Description of simulation situations.

| | Detecting Distance | Application of SDI Method |
|---|---|---|
| Situation 1 | 4–6 m | × |
| Situation 2 | 8–10 m | × |
| Situation 3 | 8–10 m | √ |

The experiment procedure is as follows:

1. First, the sensors were attached to the experiment subject (worker surrogate), and the subject stood in different positions inside the experiment area to test the effectiveness of the SDI method.

2. Next, the subject performed the 10 motions in order, with a short pause between every two motions. (The data-collecting process is performed during the construction of the dataset.)

3. Then, on the visual side, the possible human area was detected, and the depth information inside the selected area was collected. Depth optimization based on *GradientFMM* filled in empty points inside the selected area and labeled points of interest. The coordinate amplitude of points of interest was also recorded.

4. Next, on the IMU side, data changes from the four sensors were measured during the process, and a low-pass filter was employed to eliminate redundant noise.

5. Then, the acceleration changes in each sensor were used to calculate path changes by the double-integral method.

6. Changes in the points of interest, human elbows, and knees, from both the visual side and IMU side, were calculated separately to obtain the degree of change within a certain period.

7. The degree of change from both the visual and IMU sides was used to calculate the average weighting, and the result was compared with the database to find the best match.

8. Finally, the similarity from the visual side, IMU side, and sensor fusion side were compared to determine whether the sensor fusion method showed the best result.

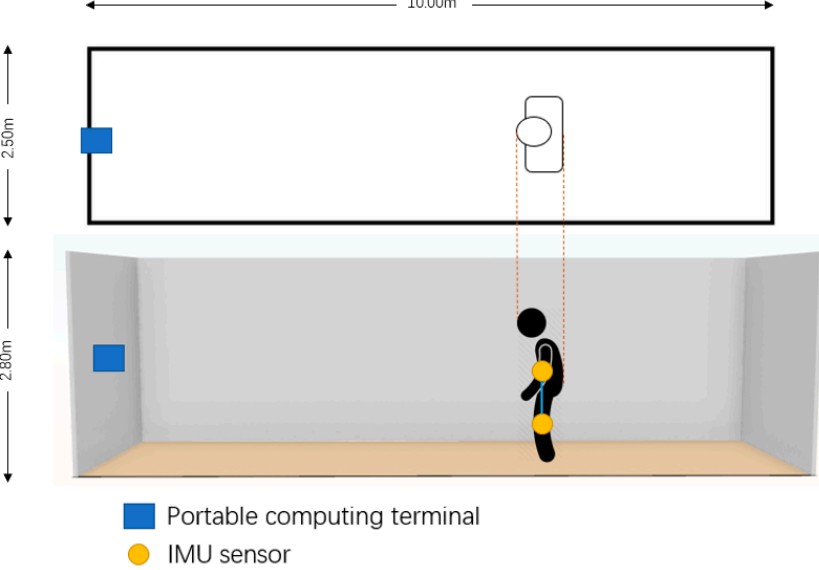

**Figure 7.** The experiment area layout (top view and 3D front view).

To process the degree of change measured by the IMUs, we set the experimental data processing environment as follows:

1. Calibration procedure: The output offset component of the acceleration sensor was removed because of the presence of static acceleration (gravity). Then, the acceleration was averaged when the accelerometer was not detecting motion (the collection of more samples improved the accuracy of the calibration result).

2. Low-pass filtering: Signal noise in the accelerometers (both mechanical and electronic) was eliminated to decrease the error while integrating the signal.

3. Mechanical filtering: When in a stationary state, small errors in acceleration were treated as constant speeds, which indicates a continuous movement and unstable position, affecting the actual motion detection. A mechanical filtering window helped to distinguish these small errors.

4. Positioning: The acceleration of each time period was known, and we used the double-integral method to obtain distance information. The first integral gain speed and the second gain distance were applied to obtain the position.

In the simulation, the 10 motions were expected to have their unique degree of change, including vector changes from the left and right elbows and knees:

$$M_n^T = \left[\vec{A_n}, \vec{B_n}, \vec{C_n}, \vec{D_n}\right], n \in [1, 10], \tag{3}$$

$$\vec{W_n} = \left(\Delta x_n^W, \Delta y_n^W, \Delta z_n^W\right), W = A, B, C, D, \tag{4}$$

where $A, B, C,$ and $D$ represent the left elbow, right elbow, left knee, and right knee, respectively, and $\vec{W_n}$ represents the change in motion from the four points of interest.

From the visual side, the collected point of interest data included pixel position and depth information, which output a vector change. From the IMU side, through the acceleration of three axes and time, Equation (1) was used to obtain the distance in all directions, thereby obtaining the vector change. Then, a weighting coefficient was assigned to the visual and IMU sides through a standard normal distribution. Next, we used Equation (2) to calculate the integrated vector change:

$$F^T = \left[\frac{\frac{A_v}{A_v^0}\Delta\alpha + \frac{A_I}{A_I^0}\Delta\beta}{2}, \frac{\frac{B_v}{B_v^0}\Delta\alpha + \frac{B_I}{B_I^0}\Delta\beta}{2}, \frac{\frac{C_v}{C_v^0}\Delta\alpha + \frac{C_I}{C_I^0}\Delta\beta}{2}, \frac{\frac{D_v}{D_v^0}\Delta\alpha + \frac{D_I}{D_I^0}\Delta\beta}{2}\right], \tag{5}$$

where $A_v$ is the left elbow vector change on the visual side, and $A_I$ is the left elbow vector change on the IMU side. $F$ is compared with $M_1$ to $M_{10}$ in Equation (3) to find the highest similarity.

Regarding the construction of the dataset, we recorded motion data from 5 male adults between the age of 20 and 30. For each motion, 50 pairs of coherent and clearly behaved data from the perspective of visual and IMU sensors are collected. This process is performed twice because the data from two different distance intervals are required. From the visual aspect, each motion is labeled based on the features such as the coordinate change in points of interest and depth information, from the IMU aspect, each motion is labeled based on the path change in each sensor attached.

This study used the same simulation method that was used in our previous paper [48]. For each motion, 100 pairs of sample data from each distance interval were prepared. The sample data were generated based on our dataset by adding random interferences and white noises, to simulate deviations caused by the effects of real data collection. Weight coefficients obeyed a standard normal distribution.

As shown in Table 1, three situations for human motion detection were considered in this simulation. To determine whether the SDI method applied to human motion detection can succeed at a relatively long distance, the results for these three situations are compared and discussed.

By combining the visual pixel changes based on the depth camera and acceleration path changes based on IMU sensors, we generated graphs with the highest similarity of each set of sample data. Some typical motion captures and their results of similarity are shown in the following figures.

Figure 8 shows the result obtained during situation 1, in which the subject stood at a highly detectable distance (around 4–6 m away from the camera), and the SDI method was not applied. The formulas for recall and precision are shown below.

$$\text{Recall}: R_{ec} = \frac{TP}{TP + FP} \tag{6}$$

$$\text{Precision}: P_{re} = \frac{TP}{TP + FN} \tag{7}$$

| Real motion / Detected motion | standing | sitting | lying down | lifting objects | picking up heavy objects | holding up heavy objects | raising arms | regular cyclical movement | bending over | kneeling | Precision |
|---|---|---|---|---|---|---|---|---|---|---|---|
| standing | 93 | 1 | | 1 | 2 | 1 | | | 1 | | 93.94% |
| sitting | | 97 | | | | | | | | | 100.00% |
| lying down | | | 98 | | | | | | | 3 | 97.03% |
| lifting objects | | | | 87 | 5 | 7 | 2 | | | | 86.14% |
| picking up heavy objects | 4 | | | 4 | 87 | 3 | | | | | 88.78% |
| holding up heavy objects | 3 | 2 | | 8 | 5 | 89 | 5 | | | | 79.46% |
| raising arms | | | | | 1 | | 93 | | | | 98.94% |
| regular cyclical movement | | | | | | | | 95 | 3 | | 96.94% |
| bending over | | | | | | | | 4 | 97 | 1 | 95.10% |
| kneeling | | | 2 | | | | | | | 96 | 97.96% |
| **Recall** | 93% | 97% | 98% | 87% | 87% | 89% | 93% | 95% | 97% | 96% | **93.27%** |

**Figure 8.** Simulation result for situation 1.

A recall is for the original sample, which indicates how many positive examples in the sample are predicted correctly. There are also two possibilities: one is to predict the original positive class as a positive class ($TP$), and the other is to predict the original positive class as a negative class ($FN$). Precision is for the prediction results, and it indicates how many of the predicted positive samples are true positive samples. Then, there are two possibilities for predicting positive instances, one is to predict the positive class as the positive class ($TP$), and the other is to predict the negative class as the positive class ($FP$).

We can learn from Figure 8 that most of the motions can be correctly identified, but in some motions, some points of interest have similar paths, which can be confused with other motions, resulting in wrong outputs. Figure 9 shows the example group of optimized depth motion captures for situation 1.

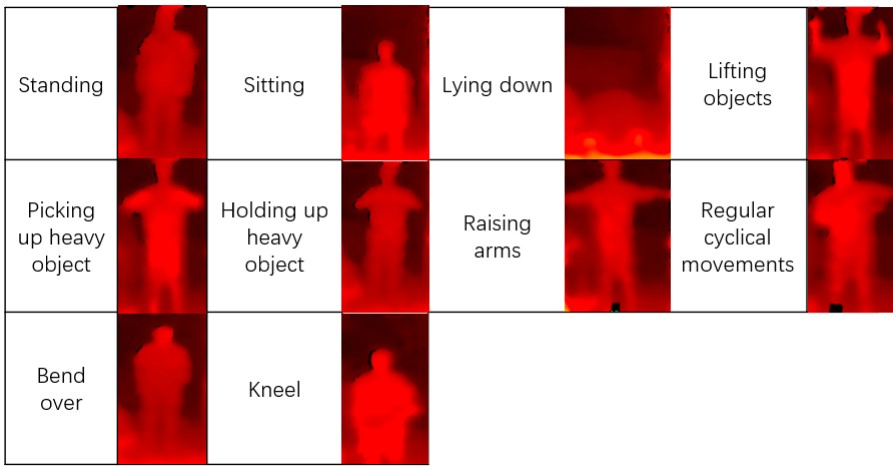

**Figure 9.** Example group of optimized depth motion capture for situation 1.

Figure 10 shows the result obtained during situation 2 when the subject stood at a relatively long distance (around 8–10 m away from the camera) and the SDI method was not applied. We can learn from Figure 11 that when people stand a long distance away from the camera, the depth detection does not perform well, especially when the detection requires the identification of similar motions with different depth changes, such as bending over, regular cyclical movement, lifting, and picking up or holding heavy things. The reason for this is that the further the object is from the depth camera, the higher the RMS error will be; even after optimization, some of the motions still cannot be identified because of errors and interference. If the fusion results are not corrected by the IMU sensor results, the accuracy could be lower.

| Real motion / Detected motion | standing | sitting | lying down | lifting objects | picking up heavy objects | holding up heavy objects | raising arms | regular cyclical movement | bending over | kneeling | Precision |
|---|---|---|---|---|---|---|---|---|---|---|---|
| standing | 90 | 2 | | 2 | 2 | 3 | 1 | | 1 | 3 | | 86.54% |
| sitting | | 84 | | | | 2 | | | 7 | 4 | 86.60% |
| lying down | | | 95 | | | | | | | | 100.00% |
| lifting objects | 3 | | | 75 | 10 | 7 | 8 | | | | 72.82% |
| picking up heavy objects | 4 | | | 15 | 65 | 19 | | | | | 63.11% |
| holding up heavy objects | 3 | 6 | | 8 | 22 | 68 | 5 | | | | 60.71% |
| raising arms | | | | | 1 | | 86 | | | | 98.85% |
| regular cyclical movement | | | | | | 1 | | 82 | 13 | | 85.42% |
| bending over | | 8 | 2 | | | | | 17 | 77 | 11 | 66.96% |
| kneeling | | | 3 | | | | | | | 85 | 96.59% |
| **Recall** | 90% | 84% | 95% | 75% | 65% | 68% | 86% | 82% | 77% | 85% | **81.07%** |

**Figure 10.** Simulation result for situation 2.

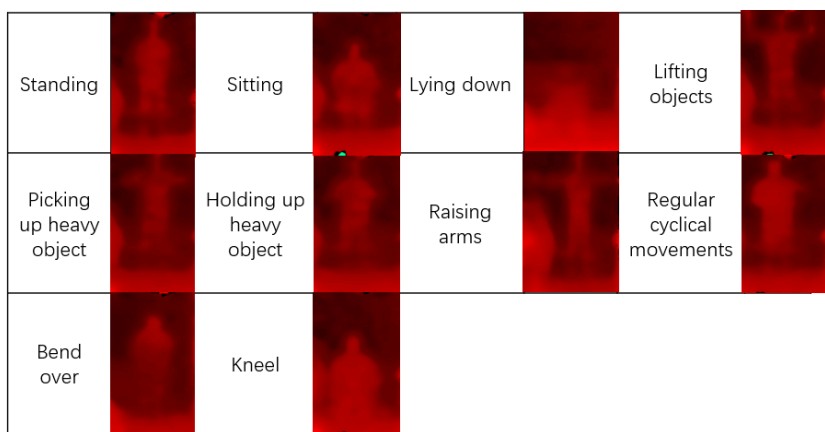

**Figure 11.** Example group of optimized depth motion capture for situation 2.

Figure 12 shows the result obtained during situation 3 when the subject stood at a relatively long distance (around 8–10 m away from the camera), but the SDI method was applied. We can learn from Figure 13 that some depth-based errors are well corrected by the SDI method using RGB person scale finding, and skeleton assistance based on PoseNet, which provide good separation for bending over and kneeling, and improve the identification between lifting and holding up or picking up heavy objects. There are still some problems while using the SDI method due to the long distance, and the skeleton or object scale is sometimes poorly constructed. In addition, the conflict between and misjudgment of some points of interest may lead the result to a completely unrelated motion, such as bending over, which was misidentified as five different motions several times.

| Real motion / Detected motion | standing | sitting | lying down | lifting objects | picking up heavy objects | holding up heavy objects | raising arms | regular cyclical movement | bending over | kneeling | Precision |
|---|---|---|---|---|---|---|---|---|---|---|---|
| standing | 98 | | | 2 | 2 | 4 | | | 2 | | 90.74% |
| sitting | | 97 | | | | | | 3 | | | 97.00% |
| lying down | | | 98 | | | | | | | 3 | 97.03% |
| lifting objects | | | | 91 | 4 | 4 | 2 | 2 | | | 88.35% |
| picking up heavy objects | 1 | | | 5 | 89 | 3 | | | 2 | | 89.00% |
| holding up heavy objects | 1 | 1 | | 2 | 3 | 89 | 3 | | 2 | | 88.12% |
| raising arms | | | | | 2 | | 95 | | 3 | | 95.00% |
| regular cyclical movement | | | | | | | | 88 | 4 | | 95.65% |
| bending over | | 2 | 1 | | | | | 7 | 87 | 1 | 88.78% |
| kneeling | | | 1 | | | | | | | 96 | 98.97% |
| **Recall** | 98% | 97% | 98% | 91% | 89% | 89% | 95% | 88% | 87% | 96% | **92.80%** |

**Figure 12.** Simulation result for situation 3.

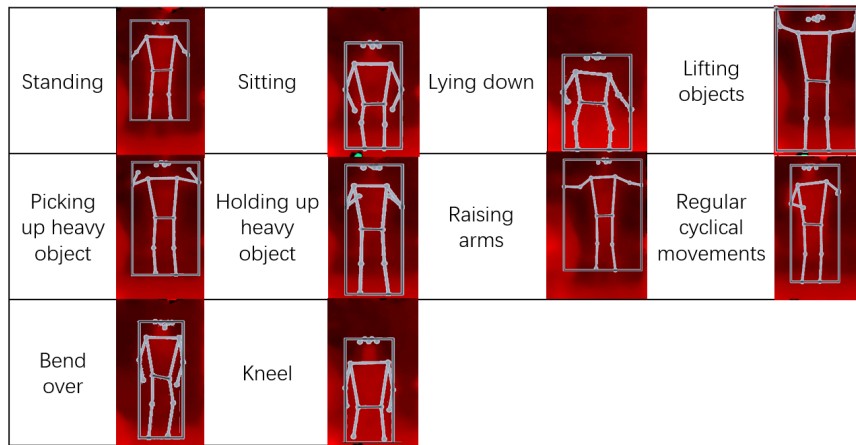

**Figure 13.** Example group of optimized depth motion capture for situation 3.

## 5. Discussion

As for the validation of the result, we considered the measure of precision and recall. Precision is based on prediction results; it indicates how many of the "predicted positive instances" are truly positive, how recall is based on original samples, and how many positive instances are predicted correctly. As the final criterion, we adopted the F1-measure approach, which is the weighted harmonic average of precision and recall. In Figure 8, the recognition result shows that the precision and recall are 93.43% and 93%, and the F1-measure is 93.27%, the similar results suggest a good classification outcome. The evaluation standard in Figures 10 and 12 is the same as in Figure 8. In Figure 10, the precision, recall, and F1-measure are 81.76%, 81%, and 81.07%, respectively; in Figure 12, the precision, recall, and F1-measure are 92.86%, 93%, and 92.80%, respectively, which also supports the good classification outcome.

A comparison of the different detection situations is shown in Figure 11, from which we can see that when the subject is far from the camera without the assistance of the SDI method, the visual side detection rate is much lower than that of the other situations. Because the Bluetooth transmission quality of the IMU sensor is also affected at long distances, the final accuracy is not very high. With the SDI method applied, long-distance motion detection can reach the accuracy of short-distance detection, and the lack of depth information is effectively compensated by image processing.

Our experiment shows that the average accuracy of motion recognition by multi-sensor fusion at a short distance (situation 1) and relatively long distance assisted by optimization from the SDI method (situation 3) can reach 93.27% and 92.80%, compared with situation 2, the accuracy improved about 12%. Although the number of samples is not large, this result shows that our proposed SDI method based on multi-sensor fusion makes it possible to realize high-precision motion recognition beyond the optimal recognition distance of the depth camera, and it can detect the different kinds of motions used at construction sites. Figure 14 showed an accuracy comparison between different detecting situations.

Currently, the assumed applicable environment of our proposed method can be described as some medium-sized indoor area of the construction site or buildings; some corners or blind spots of the construction site. Due to environmental limitations, the workers that work in a blind spot or in a closed or semi-closed area can be easily ignored, yet the danger occurrence rate of these places is very high, and because of the remote location, the rescue is not timely as well. Due to the different prerequisites, the proposed method is different from the mainstream construction site monitoring approach, and the available situations are relatively limited, but the importance is still not to be ignored.

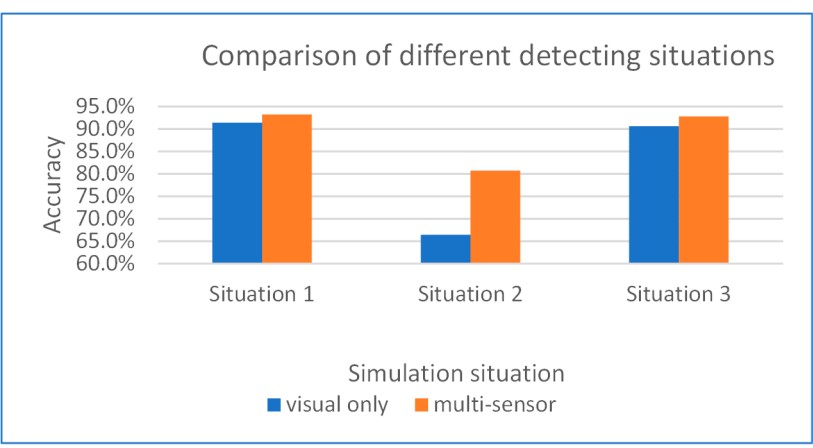

**Figure 14.** Comparison of different detection situations.

## 6. Conclusions

This paper proposes a novel method for motion recognition and hazard prevention of construction workers using an integrated sensor network. Our objective is to achieve a better observation of workers' potentially dangerous motions in dead ends and blind areas of different construction situations by monitoring workers. To effectively ensure the safety of workers in the complex environment of a construction site, we extended the detection distance of a normal depth camera. We improved the depth camera-based motion recognition with SDI, which uses preprocessing on human scale finding and depth map optimization methods to effectively reduce the detection errors and calculation burden of a broad range of depth data, while at the same time enhancing the recognition distance and accuracy of the depth camera in a selected area. We also used a portable computing terminal instead of a single depth camera to achieve local analysis, avoiding the computing burden caused by transferring a large amount of data to a central processing unit. We also demonstrated that using different types of sensors to recognize human motion improves the accuracy of motion recognition.

The proposed methodology has some limitations. To simplify the problem, we limited the considered motions in our motion recognition method to detect only 10 selected types of motion. Additionally, owing to the hardware limitations of the microcomputer, the current configuration of the SDI method cannot achieve real-time detection. The experimental data and results of the simulation were collected over time but analyzed at one time. In practical applications, data collection and analysis will be processed in a short period of time, so that even if workers experience abnormal conditions, they can be quickly discovered and rescued. At present, the result of this approach only represents its performance in a simulated environment. Although the function is basically realized, it is not yet mature enough to be applied. Our next step will be focusing on real-time realization, improving the efficiency of recognition, and the application in the real construction site environment. The results collected from an actual construction site environment are expected to have lower accuracy due to interferences.

During the simulation process, we discovered several problems, such as the points of interest for some actions having similar trajectories, resulting in some misjudgments. Therefore, finding out how to use other methods to determine and classify similar actions more accurately to improve the performance of motion recognition will be the focus of future work. Future work will also include adding a real-time warning based on motion recognition to the detection system to realize the original intention of this method, specifically, improved construction hazard prevention.

**Author Contributions:** Conceptualization, T.C.; methodology, T.C.; software, T.C.; validation, T.C.; formal analysis, T.C.; investigation, T.C.; resources, N.Y.; data curation, T.C.; writing—original draft preparation, T.C.; writing—review and editing, N.Y. and T.F.; visualization, T.C.; supervision, N.Y.

and T.F.; project administration, N.Y. and T.F.; funding acquisition, N.Y. All authors have read and agreed to the published version of the manuscript.

**Funding:** This research received no external funding.

**Data Availability Statement:** Not applicable.

**Conflicts of Interest:** The authors declare no conflict of interest.

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
