# Peer review of "Motion Recognition Method for Construction Workers Using Selective Depth Inspection and Optimal Inertial Measurement Unit Sensors"

_2673-4109, doi:10.3390/civileng4010013_

Round 1

Reviewer 1 Report

This paper introduces a multi-modal motion recognition method for construction workers using depth optimization incorporating the accelerometer. Although the paper is well-organized, there are several significant comments that have to be addressed to improve the manuscript.

ABSTRACT, BACKGROUND, and LITERATURE 

Even though the abstract describes the background of the study and the methodology, it seems that some changes need to be made to deliver the principal findings and conclusions of the paper. This reviewer recommends revising the abstract accordingly.

In lines 45 – 49, this reviewer agrees with the idea regarding the safety issues in Japan. However, more general ideas and statistics would be more appropriate to make the introduction convincing.

In lines 54 – 58, while the authors claim the reliability concern in using wearable devices, the example describes the concerns about its practicality. This reviewer does not find a connection between these statements.

In lines 88 – 94, the authors describe the disadvantages of using IMU sensors. Lastly, the authors claim that reducing the burden on workers while stabilizing detection accuracy is an issue that needs to be resolved. When this claim is made, this reviewer expects to see a way to reduce the burden of carrying sensors in a different setting. However, this reviewer found that the four IMU sensors used in the study seem to be cumbersome and their wire connections are not robust. Thus, this reviewer recommends revising the logical explanation of what the authors are trying to address with IMU sensors.

In the title and introduction, the authors mention that they proposed an advanced safety management method. However, the background information related to safety management is not described well in the paper. Which safety hazards do the authors try to address? There are lots of safety hazard categories in the construction industry. Some may be related to the worker’s motion and some maybe not. But, there is no clear explanation or description of this paper’s contribution to safety management. With the current scope, this reviewer recommends changing the title.

The literature review section needs to be further improved. Although the current literature review summarizes different methods of human modeling, more literature review is required in the field of motion recognition methods and their application in construction. This reviewer is sure that there are several recent research papers that achieved similar or higher motion recognition performance with reduced sensory infrastructures.

METHODOLOGY AND RESULT

In algorithm 1 and its description, it would be more helpful to have the full name of it when it appears the first time.

In section 3.3, the authors state that most studies on construction safety management use sensors only for information collection and they cannot provide accident alerts timely. However, this statement is somewhat bold. This reviewer can recall several research efforts for real-time safety management through different types of sensors.

In lines 303 – 311, this reviewer agrees with the strong benefits of the proposed portable computing terminal system. However, this advantage should be highlighted with evidence to make it convincing. For example, when it comes to computational efficiency, the authors can compare the time spent in a single cycle of recognition with two different approaches; centered and locally distributed.

The sensors used in the study may be able to collect various information such as temperature, height, and air pressure as the authors mentioned. However, these measures are not relevant to this paper at all. Not only they are not used in recognizing motions but also there is no description of how the authors use those measures for safety management. The author may include this in future research, but not in the other sections.

In terms of sensor fusion, the authors well-describe the procedure. However, there is no detailed information on how to determine the weight of each kind of sensor to identify the degree of change with equation 2. This is the most critical part of the proposed method because this is about how to fuse different types of sensors at the same level. The authors should provide how to determine the weight with different simulation settings.

There is a red underline in Figure 4.

For the IMU-based motion recognition module, the authors use four IMU sensors with two different specifications. It would be better to provide why the authors use the different sensors for the knees and elbows even though only tri-axial accelerations are used in the paper. If connectivity is the issue, the same sensors on the knees can be used.

The experiment design does not support the expected contribution of the proposed method well. The authors state that the existing methods cannot address the occlusion issue. However, the experiment conducted in this paper is not able to examine its robustness on occlusion. This may be critical to validate the contribution of the paper. Similarly, lines 623 – 627 are not supported by the evidence regarding blind spots or occlusion.

Lines 630 – 636 and lines 652 - 655 are not relevant to the scope of this paper. This reviewer recommends removing the statements.

Author Response

Thank you very much for your comments, please see the attachment.

Reviewer 2 Report

The paper describes about motion recognition in the construction domain, which is a unique point of the work. A sensor-fusion technique called SDI was proposed and evaluated. The application domain is interesting; however, there are a lot of concerns in the manuscript as listed below:

# General concerns:

- The manuscript should be more structured, especially Section 3, which is currently rather "flat".

- The author claims that a portable computing terminal is one of the contribution; however, it is not evaluated. Description of unused sensors such as temperature and pressure sensors would make the manuscript complicated.

- As authors described, Table 2 does not make any sense because the dataset, sensors and the recognition methods are different among the literature. 

- I could not find the parts showing the effectiveness of proposed fusion method in partial failure of wearable sensors which seems one of the advantage of the proposed method.

- The paragraphs from 389 to 399 are identical to those of 262 to 274.

# The presentation seems not following the standard writing and experimental styles of motion (activity) recognition community, which can be found in the literature of activity recognition.

- The sampling rate and the window size of the IMU signal processing was unknown. 

- It is not clear what kind of recognition model was used such as RandomForest, Support Vector Machines (SVM) and Convolutional Neural Network (CNN), was not presented. In lines 333 and 334, just saying "by comparing the new action with the label data in the database, the action with the highest similarity is output as the result. "

- Line 523 says that "several male adults" participated in the data collection. The number should be clearly presented. If it is too small, the result may get unreliable.

- The evaluation scheme was not presented. Usually, k-split cross-validation is used. For more reliable evaluation, i.e., to check the person dependency, Leave-One-Person-Out (LOPO) cross validation is often used.

- Double integration of the acceleration signal to get the distance (Eq. 1) would produce a large error by accumulating noises. So, I wonder if the effectiveness of IMU-basd recognition might be under-estimated (and the vision-based recognition might be over-estimated).

Author Response

(The authors gave the same response as above.)

Reviewer 3 Report

The article presents a solution for construction workers security. This solution is based on vision and inertial sensors.

The main issue of the article is that it is not given a context: it is not clear how the devices are located in the environment, which is the sensed/vision plane or space, what dimension it has (is it possible to monitor a whole construction area with a single camera), and most important how the feedback is achieved since the scope of the paper is safety (e.g.: how you act when you detect a dangerous situation?).

The article seems like a simple image processing report without the particularity that it is dedicated to the construction domain.

The image processing software is also not clearly presented, neither the vision architecture (besides its HW architecture - RPI4). The components of your machine vision system should be better presented (e.g.: see for example 

https://www.sciencedirect.com/science/article/pii/S2405896321007813

The subject is not new (https://www.researchgate.net/publication/362797601_Deep_Learning-Based_Automatic_Safety_Helmet_Detection_System_for_Construction_Safety) and the feedback loop to the construction workers is not obvious.

Author Response

(The authors gave the same response as above.)

Reviewer 4 Report

The paper is well-structured and written. Some improvement may be further performed. Some critique on the structure and methodology is provided here:

1)  Section 3.4 is called "IMU-based human recognition" while it described "motion recognition" and "activity recognition" in the text.

2) Figures 8, 10 and 12 represent the performance evaluation; it seems to be a confusion matrix but with the last row being "Recall"  (which values replicate the values on the diagonal of the matrix). There is also a last column showing "Precision". To be self-explanatory, the  formulae for both the recall and precision must be provided in the text.

3) Table two is title "Comparison" and uses the performance measure is called "Numerical level of accuracy" Again, the formula might be needed, and also an explanation how the accuracy for the proposed method was estimated: as an average, via cross-validation as an average or else. There is insufficient information in the text who such accuracy was calculated is a statistical sense.

4) In most cases, the paper refers to "detection accuracy", in other cases, to the "recognition accuracy". Detection and recognition are two different things. The text in sections 4 and 5 must be reviewed to avoid ambiguity of the terms.

Some typos are listed below:

1) Lines 70, 71: "... with methods such as Kinect [5] and RealSense [6]. " Note that Kinect and RealSense are not "methods" they are cameras. Overall, the word "method " is  used too often. Some of those can be replaced with "approach".

2) Line 514, a space is needed between the word "and" and vector W.

Author Response

(The authors gave the same response as above.)

Round 2

Reviewer 1 Report

Thank you for answering the comments.

Although the authors provided detailed information regarding the questions, some questions remain unsolved, and some answers did not directly address the questions.

The answer to Point 3 does not address the question properly. Does the angle detection mean a lack of reliability? Or do you mean that using only one device yields a single value and lacks reliability? For both, this review thinks the statement is not clear. Also, there are several studies that show higher performance on motion recognition with one or two IMUs. Such literature should be also included in this paper to clearly state the problem the authors are trying to address.

For Point 4, this reviewer agrees with the authors’ opinion that carrying such complex special clothing is cumbersome on a construction site. However, similar to the above question, this review thinks that the authors should conduct more literature reviews on motion recognition in construction. There are some studies showing better performance with a smaller number of sensors. This reviewer recommends conducting more literature reviews and highlighting the authors’ contribution regarding this matter.

Regarding the title, “hazard prevention” is not also relevant. This reviewer totally agrees that motion recognition can be utilized for hazard prevention. However, this paper does not have anything related to hazard prevention in it. Hazard prevention and safety management are good potential applications of this paper’s method. However, it seems not to be a good expression to be used in the title.

For Point 6, this reviewer cannot find any changes in the literature review. The response to the review should be written “after” those changes are made.

For Point 8, the authors should think about what these sentences mean because this paper is also only about information collection. As this paper’s method does not include any feedback mechanism for generating an alert to workers, the same drawback exists in the paper as well.

For Point 10, If the authors still want to demonstrate the variety of functionalities of the sensors used in the study, the authors should provide how those factors affect the data collection, experiment, and analysis as well. Currently, this reviewer can see only “At the beginning of the design of this experiment, the high temperature and the detection of potential safety hazards brought by high-altitude operations to workers were also considered, so the selection of sensors is different.” How were the temperature and altitude considered in the experiment? How do they affect motion recognition?

For Point 11, if the weights were simply determined as 50:50, the authors should explain this in the paper and mention it in the limitation section.

Similar to Point 10, in answering Point 13, this reviewer recommends providing a clear explanation of how the other information is used throughout the study.

Author Response

Thank you for the advices, I have attached the revision report, please have a look.

Reviewer 2 Report

I think the revised version fulfilled my previous concern.

Author Response

Thank you very much for your comments.

Reviewer 3 Report

Besides thanking for the comments the authors haven't implemented any of the observations. Since it is a cross-industry application (IT in construction) I shall give a final positive decision.

Author Response

Thank you very much for your comments, and thanks for the decision.